# When Cyclic Coordinate Descent Outperforms Randomized Coordinate Descent

**Mert Gürbüzbalaban**[*], **Asuman Ozdaglar**[†], **Pablo A. Parrilo**[†], **N. Denizcan Vanli**[†]
[*]Rutgers University, `mg1366@rutgers.edu`
[†]Massachusetts Institute of Technology, `{asuman,parrilo,denizcan}@mit.edu`

## Abstract

The coordinate descent (CD) method is a classical optimization algorithm that has seen a revival of interest because of its competitive performance in machine learning applications. A number of recent papers provided convergence rate estimates for their deterministic (cyclic) and randomized variants that differ in the selection of update coordinates. These estimates suggest randomized coordinate descent (RCD) performs better than cyclic coordinate descent (CCD), although numerical experiments do not provide clear justification for this comparison. In this paper, we provide examples and more generally problem classes for which CCD (or CD with any deterministic order) is faster than RCD in terms of asymptotic worst-case convergence. Furthermore, we provide lower and upper bounds on the amount of improvement on the rate of CCD relative to RCD, which depends on the deterministic order used. We also provide a characterization of the best deterministic order (that leads to the maximum improvement in convergence rate) in terms of the combinatorial properties of the Hessian matrix of the objective function.

## 1 Introduction

We consider solving smooth convex optimization problems using the coordinate descent (CD) method. The CD method is an iterative algorithm that performs (approximate) global minimizations with respect to a single coordinate (or several coordinates in the case of block CD) in a sequential manner. More specifically, at each iteration $k$, an index $i_k \in \{1, 2, \ldots, n\}$ is selected and the decision vector is updated to approximately minimize the objective function in the $i_k$-th coordinate [3, 4]. The CD method can be deterministic or randomized depending on the choice of the update coordinates. If the coordinate indices $i_k$ are chosen in a cyclic manner from the set $\{1, 2, \ldots, n\}$, then the method is called the *cyclic coordinate descent* (CCD) method. When $i_k$ is sampled uniformly from the set $\{1, 2, \ldots, n\}$, the resulting method is called the *randomized coordinate descent* (RCD) method.[1]

The CD method has a long history in optimization and its convergence has been studied extensively in 80s and 90s (cf. [5, 12, 13, 18]). It has seen a resurgence of recent interest because of its applicability and superior empirical performance in machine learning and large-scale data analysis [7, 8]. Several recent influential papers established non-asymptotic convergence rate estimates under various assumptions. Among these are Nesterov [15], which provided the first global non-asymptotic convergence rates of RCD for convex and smooth problems (see also [11, 21, 22] for problems with non-smooth terms), and Beck and Tetruashvili [1], which provided rate estimates for block coordinate gradient descent method that yields rate results for CCD with exact minimization for quadratic problems. Tighter rate estimates (with respect to [1]) for CCD are then presented in [23]. These rate estimates suggest that CCD can be slower than RCD (precisely $\mathcal{O}(n^2)$ times slower for quadratic

problems, where $n$ is the dimension of the problem), which is puzzling in view of the faster empirical performance of CCD over RCD for various problems (e.g., see numerical examples in [1, 24]). This gap was investigated in [24], which provided a quadratic problem that attains this performance gap. In this paper, we investigate performance comparison of deterministic and randomized coordinate descent and provide examples and more generally problem classes for which *CCD (or CD with any deterministic order) is faster than RCD* in terms of asymptotic worst-case convergence. Furthermore, we provide lower and upper bounds on the amount of improvement on the rate of deterministic CD relative to RCD. The amount of improvement depends on the deterministic order used. We also provide a characterization of the best deterministic order (that leads to the maximum improvement in convergence rate) in terms of the combinatorial properties of the Hessian matrix of the objective function.

In order to clarify the rate comparison between CCD and RCD, we focus on quadratic optimization problems. In particular, we consider the problem[2]

$$\min_{x \in \mathbb{R}^n} \frac{1}{2} x^T A x, \tag{1}$$

where $A$ is a positive definite matrix. We consider two problem classes: *i)* $A$ is a 2-cyclic matrix, whose formal definition is given in Definition 4.4, but an equivalent and insightful definition is the bipartiteness of the graph induced by the matrix $A - D$, where $D$ is the diagonal part of $A$; *ii)* $A$ is an M-matrix, i.e., the off-diagonal entries of $A$ are nonpositive. These matrices arise in a large number of applications such as in inference in attractive Gaussian-Markov random fields [14] and in minimization of quadratic forms of graph Laplacians (for which $A = D - W$, where $W$ denotes the weighted adjacency matrix of the graph and $D$ is the diagonal matrix given by $D_{i,i} = \sum_j W_{i,j}$), for example in spectral partitioning [6] and semisupervised learning [2]. We build on the seminal work of Young [27] and Varga [25] on the analysis of Gauss-Seidel method for solving linear systems of equations (with matrices satisfying certain properties) and provide a novel analysis that allows us to compare the asymptotic worst-case convergence rate of CCD and RCD for the aforementioned class of problems and establish the faster performance of CCD with any deterministic order.

**Outline:** In the next section, we formally introduce the CCD and RCD methods. In Section 3, we present the notion of asymptotic convergence rate to compare the CCD and RCD methods and provide a motivating example for which CCD converges faster than RCD. In Section 4, we present classes of problems for which the asymptotic convergence rate of CCD is faster than that of RCD. We provide numerical experiments in Section 5 and concluding remarks in Section 6.

**Notation:** For a matrix $H$, we let $H_i$ denote its $i$th row and $H_{i,j}$ denote its entry at the $i$th row and $j$th column. For a vector $x$, we let $x_i$ denote its $i$th entry. Throughout the paper, we reserve superscripts for iteration counters of iterative algorithms and use $x^*$ to denote the optimal solution of problem (1). For a vector $x$, $\|x\|$ denotes its Euclidean norm and for a matrix $H$, $\|H\|$ denotes its operator norm. For matrices, $\geq$ and $\leq$ are entry-wise operators. The matrices $I$ and $0$ denote the identity matrix and the zero matrix respectively and their dimensions can be understood from the context.

## 2  Coordinate Descent Method

Starting from an initial point $x^0 \in \mathbb{R}^n$, the coordinate descent (CD) method, at each iteration $k$, picks a coordinate of $x$, say $i_k$, and updates the decision vector by performing exact minimization along the $i_k$th coordinate, which for problem (1) yields

$$x^{k+1} = x^k - \frac{1}{A_{i_k,i_k}} A_{i_k} x^k e_{i_k}, \quad k = 0, 1, 2, \ldots, \tag{2}$$

where $e_{i_k}$ is the unit vector, whose $i_k$th entry is 1 and the rest of its entries are 0. Note that this is a special case of the coordinate gradient projection method (see [1]), which at each iteration updates a single coordinate, say coordinate $i_k$, along the gradient component direction (with the particular step size of $\frac{1}{A_{i_k,i_k}}$). The coordinate index $i_k$ can be selected according to a deterministic or randomized rule:

- When $i_k$ is chosen using the *cyclic rule* with order $1, \ldots, n$ (i.e., $i_k = k \pmod{n} + 1$), the resulting algorithm is called the cyclic coordinate descent (CCD) method. In order to write the CCD iterations in a matrix form, we introduce the following decomposition

$$A = D - L - L^T,$$

where $D$ is the diagonal part of $A$ and $-L$ is the strictly lower triangular part of $A$. Then, over each epoch $\ell \geq 0$ (where an epoch is defined to be consecutive $n$ iterations), the CCD iterations given in (2) can be written as

$$x_{\text{CCD}}^{(\ell+1)n} = C \, x_{\text{CCD}}^{\ell n}, \quad \text{where} \quad C = (D - L)^{-1} L^T. \tag{3}$$

Note that the epoch in (3) is equivalent to one iteration of the Gauss-Seidel (GS) method applied to the first-order optimality condition of (1), i.e., applied to the linear system $Ax = 0$ [26].

- When $i_k$ is chosen at random among $\{1, \ldots, n\}$ with probabilities $\{p_1, \ldots, p_n\}$ independently at each iteration $k$, the resulting algorithm is called the randomized coordinate descent (RCD) method. Given the $k$th iterate generated by the RCD algorithm, i.e., $x_{\text{RCD}}^k$, we have

$$\mathbb{E}_k \left[ x_{\text{RCD}}^{k+1} \mid x_{\text{RCD}}^k \right] = \left( I - SD^{-1}A \right) x_{\text{RCD}}^k, \tag{4}$$

where $S = \text{diag}(p_1, \ldots, p_n)$ contains the coordinate sampling probabilities and the conditional expectation $\mathbb{E}_k$ is taken over the random variable $i_k$ given $x_{\text{RCD}}^k$. Using the nested property of the expectations, the RCD iterations in expectation over each epoch $\ell \geq 0$ satisfy

$$\mathbb{E} x_{\text{RCD}}^{(\ell+1)n} = R \, \mathbb{E} x_{\text{RCD}}^{\ell n} \quad \text{with} \quad R := \left( I - SD^{-1}A \right)^n. \tag{5}$$

## 3 Comparison of the Convergence Rates of CCD and RCD Methods

In the following subsection, we define our basis of comparison for rates of CCD and RCD methods. To measure the performance of these methods, we use the notion of the average worst-case asymptotic rate that has been studied extensively in the literature for characterizing the rate of iterative algorithms [25]. In Section 3.2, we construct an example, for which the rate of CCD is more than twice the rate of RCD. This raises the question whether the best known convergence rates of CCD in the literature are tight or whether there exist a class of problems for which CCD provably attains better convergence rates than the best known rates for RCD, a question which we will answer in Section 4.

### 3.1 Asymptotic Rate of Converge for Iterative Algorithms

Consider an iterative algorithm with update rule $x^{(\ell+1)n} = Cx^{\ell n}$ (e.g., the CCD algorithm). The reduction in the distance to the optimal solution of the iterates generated by this algorithm after $\ell$ epochs is given by

$$\frac{\left\| x^{\ell n} - x^* \right\|}{\left\| x^0 - x^* \right\|} = \frac{\left\| C^\ell (x^0 - x^*) \right\|}{\left\| x^0 - x^* \right\|}. \tag{6}$$

Note that the right hand side of (6) can be as large as $\left\| C^\ell \right\|$, hence in the worst-case, the average decay of distance at each epoch of this algorithm is $\left\| C^\ell \right\|^{1/\ell}$. Over any finite epochs $\ell \geq 1$, we have $\left\| C^\ell \right\|^{1/\ell} \geq \rho(C)$ and $\left\| C^\ell \right\|^{1/\ell} \to \rho(C)$ as $\ell \to \infty$ by Gelfand's formula. Thus, we define the *asymptotic worst-case convergence rate* of an iterative algorithm (with iteration matrix $C$) as follows

$$\text{Rate(CCD)} := \lim_{\ell \to \infty} \sup_{x^0 \in \mathbb{R}^n} -\frac{1}{\ell} \log \left( \frac{\left\| x^{\ell n} - x^* \right\|}{\left\| x^0 - x^* \right\|} \right) = -\log \left( \rho(C) \right). \tag{7}$$

We emphasize that this notion has been used extensively for studying the performance of iterative methods such as GS and Jacobi methods [5, 18, 25, 27]. Note that according to our definition in (7), larger rate means faster algorithm and we will use these terms interchangably in throughout the paper.

Analogously, for a randomized algorithm with expected update rule $\mathbb{E} x^{(\ell+1)n} = R \, \mathbb{E} x^{\ell n}$ (e.g., the RCD algorithm), we consider the asymptotic convergence of the expected iterate error

$\left|\left|\mathbb{E}(x^{\ell n}) - x^*\right|\right|$ and define the asymptotic worst-case convergence rate as

$$\text{Rate}(\text{RCD}) := \lim_{\ell \to \infty} \sup_{x^0 \in \mathbb{R}^n} -\frac{1}{\ell} \log \left( \frac{\left|\left|\mathbb{E}(x^{\ell n}) - x^*\right|\right|}{\left|\left|x^0 - x^*\right|\right|} \right) = -\log\left(\rho(R)\right), \tag{8}$$

Note that in (8), we use the distance of the expected iterates $\left|\left|\mathbb{E}x^{\ell n} - x^*\right|\right|$ as our convergence criterion. One can also use the expected distance (or the squared distance) of the iterates $\mathbb{E}\left|\left|x^{\ell n} - x^*\right|\right|$ as the convergence criterion, which is a stronger convergence criterion than the one in (8). This follows since $\mathbb{E}\left|\left|x^{\ell n} - x^*\right|\right| \geq \left|\left|\mathbb{E}x^{\ell n} - x^*\right|\right|$ by Jensen's inequality and any convergence rate on $\mathbb{E}\left|\left|x^{\ell n} - x^*\right|\right|$ immediately implies at least the same convergence rate on $\left|\left|\mathbb{E}x^{\ell n} - x^*\right|\right|$ as well. Since we consider the reciprocal case, i.e., obtain a convergence rate on $\left|\left|\mathbb{E}x^{\ell n} - x^*\right|\right|$ and show that it is slower than that of CCD, our results naturally imply that the convergence rate on $\mathbb{E}\left|\left|x^{\ell n} - x^*\right|\right|$ is also slower than that of CCD.

### 3.2 A Motivating Example

In this section, we provide an example for which the (asymptotic worst-case convergence) rate of CCD is better than the one of RCD and building on this example, in Section 4, we construct a class of problems for which CCD attains a better rate than RCD. For some positive integer $n \geq 1$, consider the $2n \times 2n$ symmetric matrix

$$A = I - L - L^T, \quad \text{where} \quad L = \frac{1}{n^2} \begin{bmatrix} 0_{n \times n} & 0_{n \times n} \\ \mathbb{1}_{n \times n} & 0_{n \times n} \end{bmatrix}, \tag{9}$$

and $\mathbb{1}_{n \times n}$ is the $n \times n$ matrix with all entries equal to $1$ and $0_{n \times n}$ is the $n \times n$ zero matrix. Noting that $A$ has a special structure ($A$ is equal to the sum of the identity matrix and the rank-two matrix $-L - L^T$), it is easy to check that $1 - 1/n$ and $1 + 1/n$ are eigenvalues of $A$ with the corresponding eigenvectors $[\mathbb{1}_{1 \times n} \quad \mathbb{1}_{1 \times n}]^T$ and $[\mathbb{1}_{1 \times n} \quad -\mathbb{1}_{1 \times n}]^T$. The remaining $2n - 2$ eigenvalues of $A$ are equal to $1$.

The iteration matrix of the CCD algorithm when applied to the problem in (1) with the matrix (9) can be found as

$$C = \begin{bmatrix} 0_{n \times n} & \frac{1}{n^2}\mathbb{1}_{n \times n} \\ 0_{n \times n} & \frac{1}{n^3}\mathbb{1}_{n \times n} \end{bmatrix}.$$

The eigenvalues of $C$ are all zero except the eigenvalue of $1/n^2$ with the corresponding eigenvector $[n\mathbb{1}_{1 \times n}, \mathbb{1}_{1 \times n}]^T$. Therefore, $\rho(C) = 1/n^2$ and $\text{Rate}(\text{CCD}) = -\log(\rho(C)) = 2\log n$. On the other hand, the spectral radius of the expected iteration matrix of RCD can be found as

$$\rho(R) = \left( 1 - \frac{\lambda_{\min}(A)}{n} \right)^n \geq 1 - \lambda_{\min}(A) = \frac{1}{n},$$

which yields $\text{Rate}(\text{RCD}) = -\log(\rho(R)) \leq \log n$. Thus, we conclude

$$\frac{\text{Rate}(\text{CCD})}{\text{Rate}(\text{RCD})} \geq 2, \quad \text{for all} \quad n \geq 1.$$

That is, CCD is at least twice as fast as RCD in terms of the the asymptotic rate. This motivates us to investigate if there exists a more general class of problems for which the asymptotic worst-case rate of CCD is larger than that of RCD. The answer to this question turns out to be positive as we describe in the following section.

## 4 When Deterministic Orders Outperform Randomized Sampling

In this section, we present special classes of problems (of the form (1)) for which the asymptotical worst-case rate of CCD is larger than that of RCD. We begin our discussion by highlighting the main assumption we will use in this section.

**Assumption 4.1.** *A is a symmetric positive definite matrix whose smallest eigenvalue is $\mu$ and the diagonal entries of A are 1.*

Given any positive definite matrix $A$ with diagonals $D \neq I$, the diagonal entries of the preconditioned matrix $D^{-1/2}AD^{-1/2}$ are 1. Therefore, Assumption 4.1 is mild. The relationship between the smallest eigenvalue of the original matrix and the preconditioned matrix are as follows. Let $\sigma > 0$ and $L_{\max}$ denote the smallest eigenvalue and the largest diagonal entry of the original matrix, respectively. Then, the smallest eigenvalue of the preconditioned matrix satisfies $\mu \geq \sigma/L_{\max}$.

**Remark 4.2.** *For the RCD algorithm, the coordinate index $i_k \in \{1, \ldots, n\}$ (at iteration $k$) can be chosen using different probability distributions $\{p_1, \ldots, p_n\}$. Two common choices of distributions are $p_i = \frac{1}{n}$, for all $i \in \{1, \ldots, n\}$ and $p_i = \frac{A_{i,i}}{\sum_{J=1}^{N} A_{j,j}}$ [15]. Since by Assumption 4.1, the diagonal entries of $A$ are 1, both of these distributions reduces to $p_i = \frac{1}{n}$, for all $i \in \{1, \ldots, n\}$. Therefore, in the rest of the paper, we consider the RCD algorithm with uniform and independent coordinate selection at each iteration.*

In the following lemma, we characterize the spectral radius of the RCD method. This worst-case rate has been presented in many works in the literature for strongly convex optimization problems [15, 26]. The proof is deferred to Appendix.

**Lemma 4.3.** *Suppose Assumption 4.1 holds. Then, the spectral radius of the expected iteration matrix $R$ of the RCD algorithm (defined in (5)) is given by*

$$\rho(R) = \left(1 - \frac{\mu}{n}\right)^n. \tag{10}$$

## 4.1 Convergence Rate of CCD for 2-Cyclic Matrices

In this section, we introduce the class of 2-cyclic matrices and show that the asymptotic worst-case rate of CCD is more than two times faster than that of RCD.

**Definition 4.4** (2-Cyclic Matrix). *A matrix $H$ is 2-cyclic if there exists a permutation matrix $P$ such that*

$$PHP^T = D + \begin{bmatrix} 0 & B_1 \\ B_2 & 0 \end{bmatrix}, \tag{11}$$

*where the diagonal null submatrices are square and $D$ is a diagonal matrix.*

This definition can be interpreted as follows. Let $H$ be a 2-cyclic matrix, i.e., $H$ satisfies (11). Then, the graph induced by the matrix $H - D$ is bipartite. The definition in (11) is first introduced in [27], where it had an alternative name: *Property A*. A generalization of this property is later introduced by Varga to the class of $p$-cyclic matrices [25] where $p \geq 2$ can be arbitrary.

We next introduce the following definition that will be useful in Theorem 4.12 and explicitly identify the class of matrices that satisfy this definition in Lemma 4.6.

**Definition 4.5** (Consistently Ordered Matrix). *For a matrix $H$, let $H = H_D - H_L - H_U$ be its decomposition such that $H_D$ is a diagonal matrix, $H_L$ (and $H_U$) is a strictly lower (and upper) triangular matrix. If the eigenvalues of the matrix $\alpha H_L + \alpha H_U - \gamma H_D$ are independent of $\alpha$ for any $\gamma \in \mathbb{R}$ and $\alpha \neq 0$, then $H$ is said to be consistently ordered.*

**Lemma 4.6.** *[27, Theorem 4.5] A matrix $H$ is 2-cyclic if and only if there exists a permutation matrix $P$ such that $PHP^T$ is consistently ordered.*

In the next theorem, we characterize the convergence rate of CCD algorithm applied to a 2-cyclic matrix. Since $\rho(R) \geq 1 - \mu$ by Lemma 4.3, the following theorem indicates that the spectral radius of the CCD iteration matrix is smaller than $\rho^2(R)$.

**Theorem 4.7.** *Suppose Assumption 4.1 holds and $A$ is a consistently ordered 2-cyclic matrix. Then, the spectral radius of the CCD algorithm is given by*

$$\rho(C) = (1 - \mu)^2.$$

**Remark 4.8.** *Note that our motivating example in (9) is an example of a consistently ordered 2-cyclic matrix, for which Theorem 4.7 is applicable. In particular, for the example in (9), we can apply Theorem 4.7 with $\mu = 1 - 1/n$ leading to $\rho(C) = 1/n^2$, which exactly coincides with our previous computations of $\rho(C)$ in Section 3.2. We also give an example in Appendix F, for which CCD is twice faster than RCD for any arbitrary initialization with probability one.*

The following corollary states that the asymptotic worst-case rate of CCD is more than twice larger than that of RCD for quadratic problems whose Hessian is a 2-cyclic matrix. This corollary directly follows by Theorem 4.7 and definitions (7)-(8).

**Corollary 4.9.** *Suppose Assumption 4.1 holds and $A$ is a consistently ordered 2-cyclic matrix. Then, the asymptotic worst-case rates of CCD and RCD satisfy*

$$\frac{\text{Rate(CCD)}}{\text{Rate(RCD)}} = 2\nu_n, \quad where \quad \nu_n := \frac{\log(1-\mu)}{n\log\left(1-\frac{\mu}{n}\right)}. \tag{12}$$

In the following remark, we highlight several properties of the constant $\nu_n$.

**Remark 4.10.** *$\nu_n$ is a monotonically increasing function of $n$ over the interval $[1, \infty)$, where $\nu_1 = 1$ and $\lim_{n\to\infty} \nu_n = \frac{-\log(1-\mu)}{\mu} > 1$. Furthermore, $\lim_{\mu\to 0^+} \nu_n = 1$.*

We emphasize that the CCD method applied to 1 is equivalent to the Gauss-Seidel (GS) algorithm applied to the linear system $Ax = 0$ and when $A$ is a 2-cyclic matrix, the GS algorithm is twice as fast as the Jacobi algorithm [25, 27]. Hence, when $A$ is a 2-cyclic matrix and $\mu$ is sufficiently small, the RCD method is approximately as fast as the Jacobi algorithm.

## 4.2 Convergence Rate of CCD for Irreducible M-Matrices

In this section, we first define the class of M-matrices and then present the convergence rate of the CCD algorithm applied to quadratic problems whose Hessian is an M-matrix.

**Definition 4.11** (M-matrix). *A real matrix $A$ with $A_{i,j} \leq 0$ for all $i \neq j$ is an M-matrix if $A$ has the decomposition $A = sI - B$ such that $B \geq 0$ and $s \geq \rho(B)$.*

We emphasize that M-matrices arise in a variety of applications such as in belief propagation over Gaussian graphical models [14] and in distributed control of positive systems [20]. Furthermore, graph Laplacians are M-matrices, therefore solving linear systems with M-matrices (or equivalently solving (1) for an M-matrix $A$) arise in a variety of applications for analyzing random walks over graphs as well as distributed optimization and consensus problems over graphs (cf. [10] for a survey). For quadratic problems, the Hessian is an M-matrix if and only if the gradient descent mapping is an isotone operator [5, 22] and in Gaussian graphical models, M-matrices are often referred as attractive models [14].

In the following theorem, we provide lower and upper bounds on the spectral radius of the iteration matrix of CCD for quadratic problems, whose Hessian matrix is an irreducible M-matrix. In particular, we show that the spectral radius of the iteration matrix of CCD is strictly smaller than that of RCD for irreducible M-matrices.

**Theorem 4.12.** *Suppose Assumption 4.1 holds, $A$ is an irreducible M-matrix and $n \geq 2$. Then, the iteration matrix of the CCD algorithm $C = (I - L)^{-1}L^T$ satisfies the following inequality*

$$(1-\mu)^2 \leq \rho(C) \leq \frac{1-\mu}{1+\mu}, \tag{13}$$

*where the inequality on the left holds with equality if and only if $A$ is a consistently ordered matrix.*

An immediate consequence of Theorem 4.12 is that for quadratic problems whose Hessian is an irreducible M-matrix, the best cyclic order that should be used in CCD can be characterized as follows.

**Remark 4.13.** *The standard CCD method follows the standard cyclic order $(1, 2, \ldots, n)$ as described in Section 2. However, we can construct a CCD method that follows an alternative deterministic order by considering a permutation $\pi$ of $\{1, 2, \ldots, n\}$, and choosing the coordinates according to the order $(\pi(1), \pi(2), \ldots, \pi(n))$ instead. For any given order $\pi$, (1) can be reformulated as follows*

$$\min_{x_\pi \in \mathbb{R}^n} \frac{1}{2} x_\pi^T A_\pi x_\pi, \quad where \quad A_\pi := P_\pi A P_\pi^T \quad and \quad x_\pi = P_\pi x,$$

*where $P_\pi$ is the corresponding permutation matrix of $\pi$. Supposing that Assumption 4.1 holds, the corresponding CCD iterations for this problem can be written as follows*

$$x_\pi^{(\ell+1)n} = C_\pi x_\pi^{\ell n}, \quad where \quad C_\pi = (I - L_\pi)^{-1} L_\pi^T \quad and \quad L_\pi = P_\pi L P_\pi.$$

*If $A$ is an irreducible M-matrix and satisfies Assumptions 4.1, then so does $A_\pi$. Consequently, Theorem 4.12 yields the same upper and lower bounds (in (13)) on $\rho(C_\pi)$ as well, i.e., the spectral radius of the iteration matrix of CCD with any cyclic order $\pi$ satisfies*

$$(1-\mu)^2 \le \rho(C_\pi) \le \frac{1-\mu}{1+\mu}, \tag{14}$$

*where the inequality on the left holds with equality if and only if $A_\pi$ is a consistently ordered matrix. Therefore, if a consistent order $\pi^*$ exists, then the CCD method with the consistent order $\pi^*$ attains the smallest spectral radius (or equivalently, the fastest asymptotic worst-case convergence rate) among the CCD methods with any cyclic order.*

**Remark 4.14.** *The irreducibility of $A$ is essential to derive the lower bound in (13) of Theorem 4.12. However, the upper bound in (13) holds even when $A$ is a reducible matrix.*

We next compare the spectral radii bounds for CCD (given in Theorem 4.12) and RCD (given in Lemma 4.3). Since $\mu > 0$, the right-hand side of (13) can be relaxed to $(1-\mu)^2 \le \rho(C) < 1-\mu$. A direct consequence of this inequality is the following corollary, which states that the asymptotic worst-case rate of CCD is strictly better than that of RCD at least by a factor that is strictly greater than 1.

**Corollary 4.15.** *Suppose Assumption 4.1 holds, $A$ is an irreducible M-matrix and $n \ge 2$. Then, the asymptotic worst-case rates of CCD and RCD satisfy*

$$1 < \nu_n < \frac{\text{Rate}(\text{CCD})}{\text{Rate}(\text{RCD})} \le 2\nu_n, \quad \text{where} \quad \nu_n := \frac{\log(1-\mu)}{n \log\left(1-\frac{\mu}{n}\right)}, \tag{15}$$

*and the inequality on the right holds with equality if and only if $A$ is a consistently ordered matrix.*

In the following corollary, we highlight that as the smallest eigenvalue of $A$ goes to zero, the asymptotic worst-case rate of the CCD algorithm becomes twice the asymptotic worst-case rate of the RCD algorithm.

**Corollary 4.16.** *Suppose Assumption 4.1 holds, $A$ is an irreducible M-matrix and $n \ge 2$. Then, we have*

$$\lim_{\mu \to 0^+} \frac{\text{Rate}(\text{CCD})}{\text{Rate}(\text{RCD})} = 2.$$

## 5 Numerical Experiments

In this section, we compare the performance of CCD and RCD through numerical examples. First, we consider the quadratic optimization problem in (1), where $A$ is an $n \times n$ matrix defined as follows

$$A = I - L - L^T, \quad \text{where} \quad L = \frac{1}{n} \begin{bmatrix} 0 & 0 \\ \mathbb{1}_{\frac{n}{2} \times \frac{n}{2}} & 0 \end{bmatrix}, \tag{16}$$

and $\mathbb{1}_{\frac{n}{2} \times \frac{n}{2}}$ is the $\frac{n}{2} \times \frac{n}{2}$ matrix with all entries equal to 1. Here, it can be easily checked that $A$ is a consistently ordered, 2-cyclic matrix. By Theorem 4.7 and Corolloary 4.9, the asymptotic worst-case convergence rate of CCD on this example is

$$2\nu_n = 2\frac{\log(1-\mu)}{n \log\left(1-\frac{\mu}{n}\right)} = \frac{\log(0.5)}{50 \log\left(1-\frac{1}{200}\right)} \approx 2.77 \tag{17}$$

times faster than that of RCD. This is illustrated in Figure 1 (left), where the distance to the optimal solution is plotted in a logarithmic scale over epochs. Note that even if our results our asymptotic, we see the same difference in performances on the early epochs (for small $\ell$). On the other hand, when the matrix $A$ is not consistently ordered, according to Theorem 4.12, CCD is still faster but the difference in the convergence rates decreases with respect to the consistent ordering case. To illustrate this, we need to generate an inconsistent ordering of the matrix $A$. For this goal, we generate a permutation matrix $P$ and replace $A$ with $A_P := PAP^T$ in the optimization problem (1) (This is equivalent to solving the system $A_P x = 0$) so that $A_P$ is not consistently ordered (We generate $P$ randomly and compute $A_P$). Figure 1 (right) shows that for this inconsistent ordering CCD is still faster compared to RCD, but not as fast (the slope of the decay of error line in blue marker is less steep) predicted by our theory.

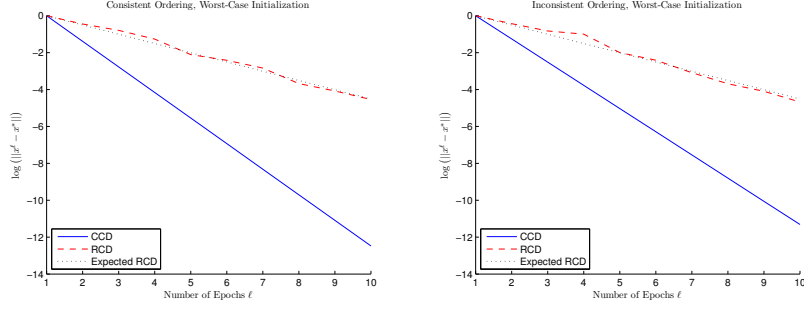

Figure 1: Distance to the optimal solution of the iterates of CCD and RCD for the cyclic matrix in (16) (left figure) and a randomly permuted version of the same matrix (right figure) where the y-axis is on a logarithmic scale. The left (right) figure corresponds to the consistent (inconsistent) ordering for the same quadratic optimization problem.

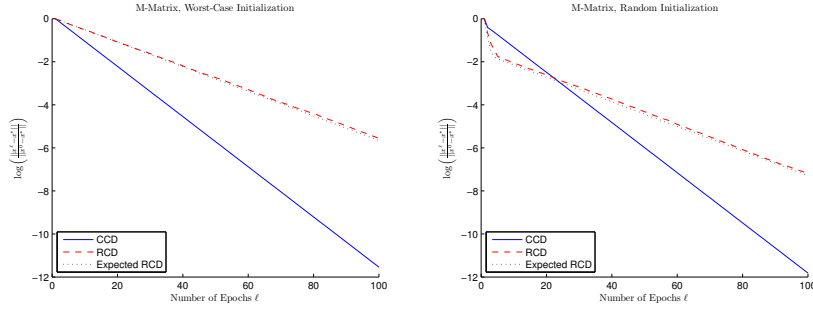

Figure 2: Distance to the optimal solution of the iterates of CCD and RCD for the M-matrix matrix in (18) for the worst-case initialization (left figure) and a random initialization (right figure).

We next consider the case, where $A$ is an irreducible positive definite M-matrix. In particular, we consider the matrix

$$A = (1 + \delta)I - \delta \mathbb{1}_{n \times n}, \tag{18}$$

where $\mathbb{1}_{n \times n}$ is the $n \times n$ matrix with all entries equal to 1 as before and $\delta = \frac{1}{n+5}$. We set $n = 100$ and plot the performance of CCD and RCD methods for the quadratic problem defined by this matrix. In Figure 2, we compare the convergence rate of CCD and RCD for an initial point that corresponds to a worst-case (left figure) and for a random choice of an initial point (right figure). We conclude that the asymptotic rate of CCD is faster than that of RCD demonstrating our results in Theorem 4.12 and Corollary 4.15.

## 6 Conclusion

In this paper, we compare the CCD and RCD methods for quadratic problems, whose Hessian is a 2-cyclic matrix or an M-matrix. We show by a novel analysis that for these classes of quadratic problems, CCD is always faster than RCD in terms of the asymptotic worst-case rate. We also give a characterization of the best cyclic order to use in the CCD algorithm for these classes of problems and show that with the best cyclic order, CCD enjoys more than a twice faster asymptotic worst-case rate with respect to RCD. We also provide numerical experiments that show the tightness of our results.

### Acknowledgments

This work is supported by NSF DMS-1723085 and DARPA Foundations of Scalable Statistical Learning grants.

## Footnotes

[1]Note that there are other coordinate selection rules as well (such as the Gauss-Southwell rule [17]). However, in this paper, we focus on cyclic and randomized rules.

[2]For ease of presentation, we consider minimization of $\frac{1}{2} x^T A x$, yet our results directly extend for problems of the type $\frac{1}{2} x^T A x - b^T x$ for any $b \neq 0$.

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
