[Supplementary Material]

# A  Supplementary Lemmas

In the following lemma, we highlight a property of nonsingular M-matrices, which we will use in the proof of Theorem 4.12.

**Lemma A.1.** *[19, Theorem 2] A is a nonsingular M-matrix if and only if $A^{-1}$ exists and $A^{-1} \geq 0$.*

We next introduce the following lemma, which is presented in variuos papers (e.g., [25, Lemma 4.12], [16, Corollary 1.2], [9, Theorem 1]) to analyze the spectral radii of nonnegative matrices.

**Lemma A.2.** *Let $B_\alpha = e^\alpha L + e^{-\alpha} L^T$, where $L \geq 0$ is a strictly lower triangular matrix and $\alpha \in \mathbb{R}$. Then, either $\rho(B_\alpha)$ is strictly log-convex in $\alpha$ with $\rho(B_\alpha) > \rho(B_0)$ for all $\alpha \neq 0$ or $\rho(B_\alpha)$ is constant for all $\alpha \in \mathbb{R}$ (i.e., $B_\alpha$ is a consistently ordered matrix).*

*Proof.* Suppose the largest eigenvalue of $B_\alpha$ has a multiplicity of 1. Then,

$$\rho(B_\alpha) = \lim_{t \to \infty} [\text{tr} \left((B_\alpha)^t\right)]^{1/t}. \tag{19}$$

In order to find the diagonal entries of $(B_\alpha)^t$, we consider the graph generated by the matrix $B_\alpha$ and define the weight of a walk as the product of the weights of the corresponding edges in the walk. We then observe that the $i$th diagonal of the matrix $(B_\alpha)^t$ can be written as the summation of weights of all closed walks of length $t$ (from the $i$th node to itself). In particular, consider a valid closed walk $w$ that contains edges $(i_s, i_{s+1})_{s=0}^{t-1}$ such that $i_0 = i_t = i$ and $[B_\alpha]_{i_s, i_{s+1}} > 0$ for all $s$. Then, we can define a symmetric walk $w'$ with edges $(i_{s+1}, i_s)_{s=0}^{t-1}$ and the $i$th diagonal entry of $(B_\alpha)^t$ contains the weights of both $w$ and $w'$ as summands. Furthermore, the weight of the walk $w$ can be written as $\phi_\alpha(w) = e^{c_w \alpha} \phi_0(w)$, for some integer $c_w$, where

$$\phi_0(w) = \prod_{s=0}^{t-1} [B_0]_{i_s, i_{s+1}}.$$

The weight of the symmetric walk $w'$ is then found by $\phi_\alpha(w') = e^{-c_w \alpha} \phi_0(w)$ since $B_0$ is symmetric. Therefore, the $i$th diagonal entry of $(B_\alpha)^t$ can be found as follows

$$[(B_\alpha)^t]_{i,i} = \sum_{\text{all valid walks } w} \frac{e^{c_w \alpha} + e^{-c_w \alpha}}{2} \phi_0(w).$$

It is easy to observe that $\cosh(c_w \alpha) = \frac{e^{c_w \alpha} + e^{-c_w \alpha}}{2}$ is a strictly log-convex function of $\alpha$ for any $c_w \neq 0$. Thus, if there exists a walk $w$ for which $c_w \neq 0$, then $\text{tr}\left((B_\alpha)^t\right)$ is a strictly log-convex function of $\alpha$ since $\phi_0(w) > 0$ for all valid walks. On the other hand, $\text{tr}\left((B_\alpha)^t\right)$ is constant in $\alpha$ if and only if $c_w = 0$ for all valid walks, which implies that the graph is bipartite since starting from an arbitrary node $i$ it is not possible to return back to node $i$ in odd number of steps. This together with (19) imply the statement of the lemma.

For the case the largest eigenvalue of $B_\alpha$ has a multiplicity of at least 2, we consider the matrix $\tilde{B}_\alpha(\epsilon) = B_\alpha + \epsilon I$, whose largest eigenvalue has a multiplicity of 1 for any $\epsilon > 0$. Using the same arguments as above, we can conclude that the statement of the lemma holds for any $\tilde{B}_\alpha(\epsilon)$ with $\epsilon > 0$ and taking the limit as $\epsilon \to 0^+$ concludes the proof of the lemma. ∎

# B  Proof of Lemma 4.3

By Assumption 4.1, $\mu > 0$ and $\text{tr}(A) = n$, which implies all eigenvalues of the matrix $A/n$ are in the interval $(0, 1)$. Therefore, we have

$$\rho(R) = \lambda_{\max}\left(\left(I - \frac{1}{n}A\right)^n\right) = \left(1 - \frac{1}{n}\lambda_{\min}(A)\right)^n = \left(1 - \frac{\mu}{n}\right)^n.$$

# C  Proof of Theorem 4.7

The eigenvalues of $C$ are the roots of the polynomial

$$\phi_C(\lambda) = \det(\lambda I - C) = 0.$$

As $I - L$ is nonsingular and $\det(I - L) = 1$, we have

$$\phi_C(\lambda) = \det(I - L)\det(\lambda I - C)$$
$$= \det(\lambda I - \lambda L - L^T)$$
$$= \sqrt{\lambda}\det\left(\sqrt{\lambda}I - \left(\sqrt{\lambda}L + \frac{1}{\sqrt{\lambda}}L^T\right)\right).$$

Therefore, if $\sqrt{\lambda}$ is an eigenvalue of the matrix $\sqrt{\lambda}L + \frac{1}{\sqrt{\lambda}}L^T$, then $\lambda$ is an eigenvalue of $C$. Furthermore, since the eigenvalues of the matrix $\sqrt{\lambda}L + \frac{1}{\sqrt{\lambda}}L^T$ are independent of $\lambda$, then $\sqrt{\lambda}$ is an eigenvalue of $L + L^T$ as well. Consequently, we have $\rho(C) = \rho^2(L + L^T) = \rho^2(I - A) = (1 - \mu)^2$.

## D  Proof of Theorem 4.12

Since $A$ is an M-matrix, $I - L$ is an M-matrix as well. Then by Lemma A.1, $(I - L)^{-1} \geq 0$, which implies $C = (I - L)^{-1}L^T \geq 0$. By the Perron-Frobenius Theorem, there exists a real eigenvalue of $C$ denoted by $\lambda$, and the corresponding unit-norm eigenvector $z \geq 0$ satisfying $\lambda = \rho(C) \geq 0$ and

$$(\lambda L + L^T)z = \lambda z. \tag{20}$$

Therefore, $\lambda$ is an eigenvalue of the matrix $\lambda L + L^T$. We then observe that $\lambda L + L^T$ is an irreducible matrix as $A$ is irreducible. Since the only nonnegative eigenvector of an irreducible nonnegative matrix is associated with the largest real eigenvalue of that matrix (by Perron-Frobenius Theorem), we conclude that

$$\lambda = \rho(\lambda L + L^T) = \sqrt{\lambda}\rho\left(\sqrt{\lambda}L + \frac{1}{\sqrt{\lambda}}L^T\right). \tag{21}$$

In order to obtain a lower bound on the right-hand side of (21), we use Lemma A.2, which characterizes the behavior of the spectral radius of the matrix in the right-hand side as $\lambda$ varies (note that $\lambda < 1$ since CCD converges linearly for $\mu > 0$, see. e.g. [18]). In particular, by Lemma A.2, we conclude that

$$\lambda \geq \sqrt{\lambda}\rho\left(L + L^T\right),$$

with equality if and only if $A$ is a consistently ordered matrix. This yields

$$\rho(C) \geq \rho^2\left(L + L^T\right) = \rho^2\left(I - A\right) = (1 - \mu)^2 \tag{22}$$

with equality if and only if $A$ is a consistently ordered matrix, which concludes the proof of the lower bound in (13). In order to obtain an upper bound on $\rho(C)$, we turn our attention back to (20) and multiply both sides by $z^T$ from the left. This yields

$$\lambda z^T L z + z^T L^T z = \lambda,$$

since $||z|| = 1$. Noting that $z^T L z = z^T L^T z$ and defining $\beta = z^T L z$, we obtain

$$\lambda = \frac{\beta}{1 - \beta}. \tag{23}$$

Since $\rho(L + L^T) = 1 - \mu$, then for any $||y|| = 1$, we have $y^T(L + L^T)y \leq 1 - \mu$. Picking $y = z$ in this inequality yields $2\beta \leq 1 - \mu$, which together with (23) imply the upper bound in (13).

## E  Proof of Corollary 4.16

By Theorem 4.12, we have the following worst-case asymptotical rate bounds for the CCD algorithm

$$-\log(1 - \mu) + \log(1 + \mu) \leq \text{Rate(CCD)} \leq -2\log(1 - \mu).$$

Dividing both sides of the above inequality by $-\log(1 - \mu)$, we obtain

$$1 - \frac{\log(1 + \mu)}{\log(1 - \mu)} \leq \frac{\text{Rate(CCD)}}{-\log(1 - \mu)} \leq 2.$$

Taking limit of both sides as $\mu \to 0^+$ yields

$$\lim_{\mu \to 0^+} \frac{\text{Rate(CCD)}}{-\log(1-\mu)} = 2. \tag{24}$$

By Lemma 4.3, we have the following worst-case asymptotical rate for the RCD algorithm

$$\text{Rate(RCD)} = -n \log\left(1 - \frac{\mu}{n}\right).$$

Dividing both sides of the above inequality by $-\log(1-\mu)$ and taking limit of both sides as $\mu \to 0^+$, we get

$$\lim_{\mu \to 0^+} \frac{\text{Rate(RCD)}}{-\log(1-\mu)} = 1. \tag{25}$$

Combining (24) and (25) concludes the proof.

## F  Example Achieving Lower and Upper Bounds

Consider solving the linear system $Ax = 0$ where A is defined as follows

$$A = \begin{bmatrix} 1 & -\delta \\ -\delta & 1 \end{bmatrix}$$

for some $\delta \in (0,1)$. The CCD algorithm applied to this problem has the following iteration matrix

$$C = \begin{bmatrix} 0 & \delta \\ 0 & \delta^2 \end{bmatrix},$$

whereas the expected RCD iteration matrix is

$$R = \left(I - \frac{A}{2}\right)^2 = \begin{bmatrix} 1/2 & \delta/2 \\ \delta/2 & 1/2 \end{bmatrix}^2 = \frac{1}{4}\begin{bmatrix} 1+\delta^2 & 2\delta \\ 2\delta & 1+\delta^2 \end{bmatrix}.$$

The eigendecomposition of this matrix can be found as follows

$$R = \begin{bmatrix} \frac{1}{\sqrt{2}} & -\frac{1}{\sqrt{2}} \\ \frac{1}{\sqrt{2}} & \frac{1}{\sqrt{2}} \end{bmatrix} \begin{bmatrix} \frac{1+\delta}{2} & 0 \\ 0 & \frac{1-\delta}{2} \end{bmatrix} \begin{bmatrix} \frac{1}{\sqrt{2}} & -\frac{1}{\sqrt{2}} \\ \frac{1}{\sqrt{2}} & \frac{1}{\sqrt{2}} \end{bmatrix}^{-1}.$$

Therefore, after $\ell$ epochs the distance of the iterates generated by RCD starting from the initial point $x^0 = [a,b]^T$ becomes

$$\mathbb{E}\left\|x^\ell - x^*\right\| = \mathbb{E}\left\|x^\ell\right\| \geq \left\|\mathbb{E}x^\ell\right\| = \left\|R^\ell x^0\right\| = \left\| \begin{bmatrix} \left(\frac{1+\delta}{2}\right)^\ell a \\ \left(\frac{1-\delta}{2}\right)^\ell b \end{bmatrix} \right\|$$

$$= \sqrt{\left(\frac{1+\delta}{2}\right)^{2\ell} a^2 + \left(\frac{1-\delta}{2}\right)^{2\ell} b^2}.$$

$$\geq \left(\frac{1+\delta}{2}\right)^\ell |a|$$

$$\geq \delta^\ell |a|.$$

Therefore, in order to achieve a solution in the $\epsilon$-neighborhood of the optimal solution $x^* = 0$, i.e., to attain $\left\|x^\ell - x^*\right\| = \epsilon$, the RCD method requires

$$N_R(\epsilon) \geq \frac{\log \epsilon}{\log \delta} - \frac{\log |a|}{\log \delta}$$

epochs, for any $a \neq 0$.

On the other hand, for the CCD algorithm, we have

$$C^\ell = \begin{bmatrix} 0 & \delta^{2\ell-1} \\ 0 & \delta^{2\ell} \end{bmatrix},$$

and consequently the suboptimality of the iterates generated by the CCD algorithm is

$$\left\lVert C^{\ell} x_0 \right\rVert = \delta^{2\ell} \sqrt{b^2 + \frac{1}{\delta^2} b^2}.$$

Therefore, in order to achieve a solution in the $\epsilon$-neighborhood of the optimal solution $x^* = 0$, i.e., to attain $\left\lVert x^{\ell} - x^* \right\rVert = \epsilon$, the CCD method requires

$$N_C(\epsilon) = \frac{\log \epsilon}{2 \log \delta} - \frac{\log \left( b^2 + \frac{1}{\delta^2} b^2 \right)}{4 \log \delta}$$

epochs.

Note that for small $\epsilon$ the first terms in the expression of $N_J(\epsilon)$ and $N_C(\epsilon)$ are dominant. In particular we have,

$$\lim_{\epsilon \to 0^+} \frac{N_R(\epsilon)}{N_C(\epsilon)} => \frac{2 \log \delta}{\log \delta} = 2, \tag{26}$$

for any $a \neq 0$.