[Reviews · NeurIPS 2017]

Reviewer 1



This paper gives nice examples to show that cyclic coordinate descent can be asymptotically faster than randomized coordinate descent. Though different to those existing non-asymptotic iteration complexity analysis, this paper is still of certain interest for nonlinear optimization. I therefore recommend this paper to be accepted. My detailed comments are as follows. 1. The claim that people have the perception that CCD is always inferior is false. At least in the case that n is small, this can be not the case. The paper by Sun and Ye only claimed that it scales worse with respect to n. [6] also provided an example that random permutation CD (which usually behaves closer to RCD than to CCD) can be worse than CCD when n = 2. 2. In line 110: the result for the case l approaches infinity is called Gelfand's formula (as mentioned in [11]). 3. I would still hope to see the rate of RCD represented in E || x - x^* ||, as this value can be directly related to the expected objective value, which is the one that is really of interest. As noted in [20], the current analysis for RCD cannot be easily linked with the expected objective and therefore the result is weaker. It should not be too difficult to obtain such results by utilizing Nesterov's results in [14]. 4. I am somehow confused by the notations: in definition 4.3, is the inequality element-wise or is it for the eigenvalues? In the notation section it is claimed that the norm is always the Euclidean norm, but there are also norms applied on the matrices, which clearly cannot be the Euclidean norm. 5. [17] and [18] are referring to the same paper, and the correct papers to refer should be "Richtárik, Peter, and Martin Takáč. "Iteration complexity of randomized block-coordinate descent methods for minimizing a composite function." Mathematical Programming 144.1-2 (2014): 1-38." and "Lu, Zhaosong, and Lin Xiao. "On the complexity analysis of randomized block-coordinate descent methods." Mathematical Programming 152.1-2 (2015): 615-642." 6. In the outline section, description of sec. 4 was ignored. 7. In the notation section, the part about the superscripts is mentioned twice. 8. The D matrix was used in line 51 without a definition, and later it's mentioned that the diagonal parts of A is denoted by I while in (4) D is used. Though under the assumption that all diagonal entries are 1, they are equivalent, it is still better to unify the notations. === After reading the authors' feedback === I thank the authors for the feedback and have no further comments.

Reviewer 2



In this paper, the authors analyze cyclic and randomized coordinate descent and show that despite the common assumption in the literature, cyclic CD can actually be much faster than randomized CD. The authors show that this is true for quadratic objectives when A is of a certain type (e.g., symmetric positive definite with diagonal entries of 1, irreducible M-matrix, A is consistently ordered, 2-cyclic matrix). Comments: - There are more than 2 variants of CD selection rules: greedy selection is a very valid selection strategy for structured ML problems (see "Coordinate Descent Converges Faster with the Gauss-Southwell Rule than Random Selection, ICML 2015 by Nutini et. al.) - The authors state that it is common perception that randomized CD always dominates cyclic CD. I don't agree with this statement. For example, if your matrix is diagonal, cyclic CD will clearly do better than randomized CD. - The authors make several assumptions on the matrices considered in their analysis (e.g., M-matrix, 2-cyclic). It is not obvious to me that these matrices are common in machine learning applications when solving a quadratic problem. I think the authors need to do a better job at convincing the reader that these matrices are important in ML applications, e.g., precision matrix estimation https://arxiv.org/pdf/1404.6640.pdf "Estimation of positive definite M-matrices and structure learning for attractive Gaussian Markov Random fields" by Slawski and Hein, 2014. - The authors should be citing "Improved Iteration Complexity Bounds of Cyclic Block Coordinate Descent for Convex Problems" NIPS 2015 (Sun & Hong) - I have several issues with the numerical results presented in this paper. The size of the problem n = 100 is small. As coordinate descent methods are primarily used in large-scale optimization, I am very curious why the authors selected such a small system to test. Also, it seems that because of the structure of the matrices considered, there is equal progress to be made regardless of the coordinate selected -- thus, it seems obvious that cyclic would work better than random, as random would suffer from re-selection of coordinates, while cyclic ensures updating every coordinate in each epoch. In other words, the randomness of random selection would be unhelpful (especially using a uniform distribution). Can the authors please justify these decisions. Overall, I'm not convinced that the analysis in this paper is that informative. It seems that the assumptions on the matrix naturally lend to using cyclic selection as there is equal progress to be made by updating any coordinate. At least that is what the numerical results are showing. I think the authors need to further convince that this problem setting is important and realistic to ML application and that their numerical results emphasize again the *realistic* benefits of cyclic selection. ============ POST REBUTTAL ============ I have read the author rebuttal and I thank the authors for their details comments. My comment regarding "equal progress to be made" was with respect to the structure in the matrix A and the authors have address this concern, pointing my attention to the reference in [20]. I think with the inclusion of the additional references mentioned by the authors in the rebuttal that support the applicability of the considered types of matrices in ML applications, I can confidently recommend this paper for acceptance.

Reviewer 3



This paper proved that for two classes of matrices, cyclic coordinate descent (CCD) is asymptotically faster than randomized coordinate descent (RCD). In particular, they consider two classes of matrices: M-matrices and matrices with Young’s property A (called 2-cyclic in the paper), and prove that the asymptotic convergence rate of CCD is twice as fast as RCD. The comparison of CCD and RCD is an interesting problem. While the recent work [20] proved in the worst case CCD can be O(n^2) times slower, it remains an interesting question why in many practical problems CCD can be faster than or comparable to RCD. This paper gives one of the first positive results for CCD, for some special classes. These results shed light on the understanding of CCD and RCD. To improve the paper, I have a few suggestions: (i) In line 34, the authors said that “these rate estimates suggest that CCD can be slower (precisely O(n^2) times slower)“. This is not “precise” to my knowledge. The rate estimate of [1] for CCD actually can be O(n^3) times slower than RCD. This gap was shrunk to O(n^2) for quadratic problems in [Sun, Hong ’15 NIPS], but the O(n^3) gap for the general convex problem was still there (though partially reduced for some problems in that paper). In [20], the gap for general convex problem was mentioned as an open question. (ii) “The perception that RCD always dominates CCD” (line 37) was mentioned in [6], but probably not “in light of this result“ of [20]. In fact, [20] did give numerical examples that CCD is faster than RCD: for matrices with standard Gaussian entries, CCD is indeed faster than RCD in the simulation. [20] also mentioned that the worst-case analysis does not mean CCD is bad, and other analysis methods are needed to explain why CCD is usually faster than GD. I think at least the numerical example of Gaussian case should be mentioned in the paper. (iii) It is well known that when Young’s property A holds, CCD is twice as fast as Jacobi method. So for the 2-cyclic case, the Jacobi method is approximately as fast as RCD, right? Could the authors explain the connection between Corollary 4.16 and the classical result (or claim it is just a coincidence)? Overall, I think this is a good paper that proved some interesting results. Only a few minor issues regarding the literature need to be explained before acceptance.